# Inhibition of extracellular signal-regulated kinase pathway suppresses tracheal stenosis in a novel mouse model

**Akari Kimura**[1], **Koji Araki**[2]*, **Yasushi Satoh**[3], **Sachiyo Mogi**[1], **Kazuko Fujitani**[4],
**Takaomi Kurioka**[1], **Shogo Endo**[5], **Akihiro Shiotani**[2], **Taku Yamashita**[1]

**1** Department of Otorhinolaryngology-Head and Neck Surgery, Kitasato University School of Medicine,
Sagamihara, Kanagawa, Japan, **2** Department of Otolaryngology-Head and Neck Surgery, National Defense
Medical College, Tokorozawa, Saitama, Japan, **3** Department of Biochemistry, National Defense Medical
College, Tokorozawa, Saitama, Japan, **4** Department of Gene Analysis Center, Kitasato University School of
Medicine, Sagamihara, Kanagawa, Japan, **5** Aging Neuroscience Research Team, Tokyo Metropolitan
Geriatric Hospital and Institute of Gerontology, Itabashi, Tokyo, Japan

* kojaraki@ndmc.ac.jp

org/10.1371/journal.pone.0256127

TAIWAN

**Data Availability Statement:** All relevant data are
within the manuscript and its Supporting
Information files.

## Abstract

Tracheal stenosis is a refractory and recurrent disease induced by excessive cell prolifera-
tion within the restricted tracheal space. We investigated the role of extracellular signal-reg-
ulated kinase (ERK), which mediates a broad range of intracellular signal transduction
processes in tracheal stenosis and the therapeutic effect of the MEK inhibitor which is the
upstream kinase of ERK. We histologically analyzed cauterized tracheas to evaluate steno-
sis using a tracheal stenosis mouse model. Using Western blot, we analyzed the phosphory-
lation rate of ERK1/2 after cauterization with or without MEK inhibitor. MEK inhibitor was
intraperitoneally injected 30 min prior to cauterization (single treatment) or 30 min prior to
and 24, 48, 72, and 96 hours after cauterization (daily treatment). We compared the stenosis
of non-inhibitor treatment, single treatment, and daily treatment group. We successfully
established a novel mouse model of tracheal stenosis. The cauterized trachea increased
the rate of stenosis compared with the normal control trachea. The phosphorylation rate of
ERK1 and ERK2 was significantly increased at 5 min after the cauterization compared with
the normal controls. After 5 min, the rates decreased over time. The daily treatment group
had suppressed stenosis compared with the non-inhibitor treatment group. p-ERK1/2 acti-
vation after cauterization could play an important role in the tracheal wound healing process.
Consecutive inhibition of ERK phosphorylation is a potentially useful therapeutic strategy for
tracheal stenosis.

## Introduction

Tracheal stenosis is a life-threatening airway narrowing disorder causes breathing difficulties
and suffocation. Tracheal stenosis is a chronic fibroproliferative disease characterized by a sec-
ondary inflammatory response that promotes increased fibroblast activity and collagen

**Funding:** A.K. Grants-in-Aid for Early-Career Scientists (No.40623137) K.A. Grant-in-Aid for Scientific Research (C) (No.18K09390, 17K11415, 16K11252) and Grant-in-Aid for Challenging Exploratory Research (No.25670723) from the Ministry of Education, Culture, Sports, Science and Technology, Japan and grants from the National Defense Medical College Special Research Grant. The funders had no role in study design, data collection and analysis, decision to publish, or preparation of the manuscript.

**Competing interests:** The authors have declared that no competing interests exist.

deposition. Acquired tracheal stenosis commonly occurs due to prolonged intubation, laser surgery, radiotherapy, and airway burns. Previous studies have reported that 5%-24% of patients with inhalational burn injuries require endotracheal intubation or tracheostomy in the acute phase [1–3]. However, endotracheal intubation and tracheostomy can induce prolonged laryngotracheal stenosis [4–6]. Tracheal laser surgery to remove the granulation tissue can worsen the stenosis caused by new scar formation. Recurrent and prolonged tracheal stenosis following surgery has been a longstanding problem and compromises the quality of life of these patients.

Wound healing consists of an inflammatory phase, a proliferative phase, and a remodeling phase [7]. Fibroblasts play a crucial role in tissue repair, starting in the late inflammatory phase by secreting growth factors, cytokines, collagens, and other extracellular matrix components. At the same time, fibroblast migration and proliferation play crucial roles by initiating the proliferative phase [8, 9]. Inhibition of the pathway to fibroproliferation during the proliferative phase might suppress tracheal stenosis.

Extracellular signal-regulated kinase (ERK), a member of the mitogen-activated protein kinase (MAPK) family regulates cell proliferation, differentiation, and apoptosis in response to a variety of external stresses [10]. MAPK/ERK kinase (MEK) is the upstream kinase of ERK and activates ERK through the phosphorylation of tyrosine and threonine residues. Although the specific role of ERK *in vivo* has not been fully determined, recent evidence suggests that the ERK pathway plays an important role in the wound healing process [11]. We therefore hypothesized that the inhibition of ERK phosphorylation is a potential strategy for treating tracheal stenosis. The purpose of this study was to inhibit excessive fibroproliferation during the proliferative phase by inhibiting ERK phosphorylation.

## Materials and methods

All protocols for the handling and experimental use of animals were approved by the Committee on the Ethics of Animal Experiments of the Kitasato University School of Medicine, Kanagawa, Japan (Permit numbers: 2019–104, 2020–072).

### 1. Animal procedures

All surgeries were performed with the animals under general anesthesia. The study used 70 male C57BL/6J mice between 7 and 9 weeks of age (20–23 g), which were anesthetized by an intraperitoneal injection of 3 types of mixed anesthetic agents: medetomidine hydrochloride (Domitol®, 0.75 mg/kg), midazolam (Dormicum®, 4 mg/kg), and butorphanol (Vetophale®, 5 mg/kg). The calculation of sample size was based on law of diminishing return, which was called "resource equation" method. By calculation, 4 mice each group were considered as enough sample size.

**1–1. Establishment of the novel mouse model of tracheal stenosis.** Twelve of the mice were assigned to three groups (n = 4 for each group): cauterized, non-cauterized (normal control) and fake surgery. The cauterized group underwent a vertical anterior neck incision after a local injection of xylocaine (1%, 50 mg/kg). We performed tracheostomy between the fourth and fifth tracheal rings and cauterized the third and fourth rings. The tip of an electric scalpel bent 90˚ was inserted through the tracheostomy. The anterior tracheal wall above the tracheostoma was cauterized for 0.5 s with the scalpel (Bovie, 1200˚C, Symmetry Surgical, TN) (Fig 1A–1C). The tracheostoma was kept open and carefully treated to prevent suffocation. Seven days after the cauterization, the mice were euthanized. Then, we removed the tracheal rings and analyzed at during the first and second rings where showed most severe tracheal stenosis. The normal control group did not undergo the surgical procedure but did have the two

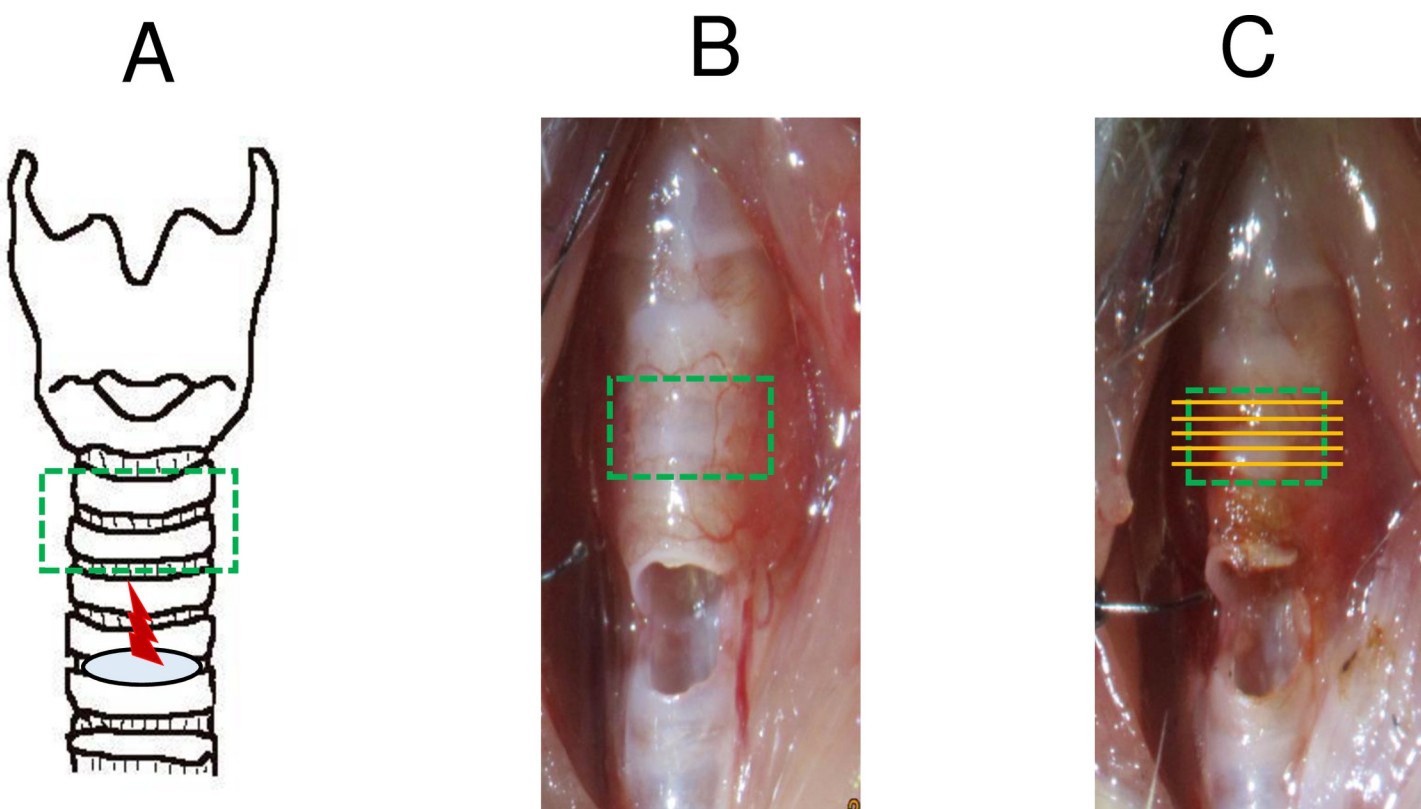

**Fig 1. Establishment of the novel mouse model of tracheal stenosis.** (A) Tracheostomy was performed between the fourth and fifth tracheal rings (blue area). The anterior tracheal wall above the tracheostoma was cauterized using an electric scalpel (red area). (B) Trachea after tracheostomy. (C) Trachea after cauterization. Green box: removed area, yellow lines: sliced sections.

tracheal rings removed after being euthanized. The fake surgery group underwent only tracheostomy and did have the two tracheal rings removed after being euthanized. The excised tracheas were embedded in paraffin for histological examination to evaluate the rate of airway stenosis, as described in Section *2. Histological evaluation.* Animals were observed once a day and euthanized when marked wheezing, and significant decrease in body movement were observed.

**1–2. P-ERK expression in the normal control and cauterized trachea.** To investigate the localized expression of phosphorylated ERK (p-ERK), we performed immunohistochemistry on the normal control trachea and cauterized trachea (n = 1 for each group). The control mice did not undergo the procedure after being administered general anesthesia but did have the two tracheal rings removed after being euthanized. The cauterized mice underwent the procedure and the two tracheal rings were removed 5 min after the cauterization. The excised trachea was embedded in paraffin, and tracheal sections were stained with p63 (basal cell marker) and p-ERK, as described in Section *3. Immunohistochemistry* [12].

**1–3. P-ERK expression over time in the cauterized tracheas.** To analyze the time-dependent change in p-ERK expression in the cauterized group, we removed two tracheal rings above the tracheostoma at 5, 30, and 90 min after cauterization (n = 4 for each group). We also removed two tracheal rings from the normal control group (n = 4). We measured the ERK1/2 and p-ERK1/2 expression of these tracheas, as described in Section *4. Western blot.*

**1–4. Inhibitory effect of ERK phosphorylation by MEK inhibitor in the normal control tracheas.** We examined the progression of ERK1/2 and p-ERK1/2 expression after the

administration of MEK inhibitor ((α-[amino[(4-aminophenyl)thio]methylene]-2-(trifluoro-methyl) benzene acetonitrile (SL-327), BML-EI1365, ENZO Life Sciences, NY) in the normal control group. The mice in the MEK inhibitor-treated normal control group were intraperito-neally injected with MEK inhibitor (50 mg/kg, dissolved in 100% dimethyl sulfoxide [DMSO]; 13445–74, Nacalai tesque, Kyoto, Japan) [13]. Two tracheal rings were removed at 30, 60, 90, and 120 min after the injection (n = 1 for each group). The expression of ERK1/2 and p-ERK1/2 in these tracheas was measured as described in Section *4. Western blot*.

**1–5. Inhibitory effect of ERK phosphorylation by MEK inhibitor in the cauterized tra-cheas.** We examined the progression of ERK1/2 and p-ERK1/2 expression after the administration MEK inhibitor in the cauterized group. The mice were injected intraperitoneally with MEK inhibitor (MEKi-treated cauterized group; 50 mg/kg, dissolved in 100% DMSO) or the same dose of DMSO (non-treated cauterized group) 30 min prior to cauterization. The tracheas of both groups were removed at 5, 30, and 90 min after cauterization (n = 2 for each group). We measured the expression of ERK1/2 and p-ERK1/2 of these tracheas as described in Section *4. Western blot*.

**1–6. Preventive effect of MEK inhibitor on tracheal stenosis.** We assigned twelve mice to three groups: non-inhibitor treatment, single treatment, and daily treatment (n = 4 for each group). All groups underwent tracheal cauterization, as described in Section *1–1*. Non-inhibi-tor treatment group was injected DMSO 30 min prior to cauterization and 24, 48, 72, and 96 h after cauterization (5 consecutive days). The single treatment group was intraperitoneally injected MEK inhibitor (50 mg/kg, dissolved in 100% DMSO) 30 min prior to cauterization. The daily treatment group was intraperitoneally injected MEK inhibitor (50 mg/kg, dissolved in 100% DMSO) 30 min prior to cauterization and 24, 48, 72, and 96 h after cauterization (5 consecutive days).

Seven days after the cauterization, we excised the tracheas of all groups and embedded the tracheas in paraffin for histological examination to evaluate the rate of airway stenosis, as described in the next section.

## 2. Histological evaluation

We fixed the excised tracheal tissues in 4% formalin, embedded them in paraffin, and cut them into 4-μm-thick axial sections. We performed hematoxylin and eosin (HE) staining and Mas-son trichrome (MT) staining according to standard procedures. We took 140-μm-thick sec-tions from the bottom of the trachea to the top, taking an average of 5 sections. We determined the rate of airway stenosis by calculating the cross-sectional area using Image J software (version 1.52u, National Institutes of Health, Bethesda, MD) using the following for-mula: (1—area of the mucosal surface lumen / area of the tracheal cartilage lumen) × 100 [14].

## 3. Immunohistochemistry

We histologically analyzed the tracheas using 4-μm-thick paraffin-embedded sections, which were dewaxed in UI sol (UI Kasei) and hydrated using a graded series of ethanol. Antigenic retrieval was performed by immersing mounted tissue sections in antigen unmasking solution and heating them in an autoclave at 121 ˚C for 5 min. The deparaffinized sections were then blocked with a nonspecific staining blocking reagent (Dako, Agilent) for 1 h to reduce back-ground staining.

The sections were then incubated overnight at 4 ˚C with one of the following primary anti-bodies: anti-p-ERK (#4370, rabbit, monoclonal, 1:400, Cell Signaling Technology) and anti-p63 (#ab735, mouse, 1:100, Abcam). We washed the sections 3 times in PBS and incubated them with a corresponding secondary antibody (Alexa Fluor 488nm [anti-mouse, 1:200] or

546nm [anti-rabbit, 1:200], IgG, Invitrogen) diluted in an antibody diluent (Dako, Agilent). We obtained images of the immunolabeled specimens using confocal laser microscopy (LSM710, Zeiss, Jena, Germany).

For 3,3′-diaminobenzidine (DAB) staining, tracheal sections were blocked with a nonspecific staining blocking reagent (Dako, Agilent) for 1 h followed by overnight incubation with anti-pERK1/2 primary antibody (#4370, rabbit, monoclonal, 1:400, Cell Signaling Technology). After extensive washing, sections were incubated with secondary antibodies (DAKO, Dual Link System-HRP) for one hour. Immunoreactivity was visualized using DAB as substrate. Sections were dehydrated with alcohol, clarified with UI sol, and cover-slipped using Neo-Mount (Merck Millipore). The stained sections were observed using an Olympus CX43 microscope with an Olympus DP22 camera.

## 4. Western blot analysis

The specimens were cut and homogenized in RIPA Buffer with Protease Inhibitor Cocktail (Nacalai tesque, Kyoto, Japan). The homogenate was then centrifuged at 15,000 rpm for 10 min at 4 ˚C. We separated the supernatants of the homogenates using sodium dodecyl sulfate polyacrylamide gel electrophoresis (12.5% e-PAGEL-HR, ATTO) and transferred the proteins onto an Immobilon-P membrane (Clear Blot Membrane-P plus, ATTO). We blotted the membranes with anti-ERK1/2 (#9102, rabbit, polyclonal, 1:1000, Cell Signaling Technology), anti-pERK1/2 (#4370, rabbit, monoclonal, 1:500, Cell Signaling Technology), or anti-β-actin (G043, mouse, 1:1000, abm) antibodies. We treated the sections with a secondary antibody (EnVision+/HRP, Dual Link Rabbit/Mouse, HRP) and visualized the protein bands with a chemiluminescence detection system (Amersham ECL Prime Western Blotting Detection Reagent). We analyzed the signals in the immunoblots using an LAS-4000 digital imaging system (Fujifilm, Tokyo, Japan). We quantified and divided the ERK and p-ERK band intensities by their corresponding loading controls with Image J software (version 1.52u) and determined the ERK phosphorylation rate by p-ERK/ERK.

## 5. Statistical analysis

The statistical analysis was performed using Graph Pad Prism 9.1.2 (Graph Pad Software Inc., La Jolla, CA). To evaluate the differences in the rate of tracheal stenosis, and to evaluate the difference in the phosphorylation rate of ERK1/2 between the groups, we analyzed the intergroup comparisons using the non-parametric Kruskal–Wallis test, followed by Dunn's multiple comparison test, with the statistical significance level set at $p$ values < 0.05.

# Results

## 1. Establishing the novel mouse model of tracheal stenosis

The HE sections of the excised trachea revealed that at 7 days after cauterization, the cauterized group showed a narrowed lumen caused by a thickened submucosal layer causing severe stenosis compared with the normal control group and the fake surgery group (Fig 2A and 2B). The MT-stained sections revealed extensive fibrosis, thickening, and collagen deposition, especially in the submucosal layer in the cauterized group.

The median rate of airway stenosis was as follows; normal control: 10.7%, fake surgery: 13.5%, cauterized: 48.2%. The cauterized group showed a significantly higher rate of airway stenosis compared with the normal control group (cauterized vs. normal control; $p = 0.007$, cauterized vs. fake surgery, $p = 0.286$) (Fig 2C).

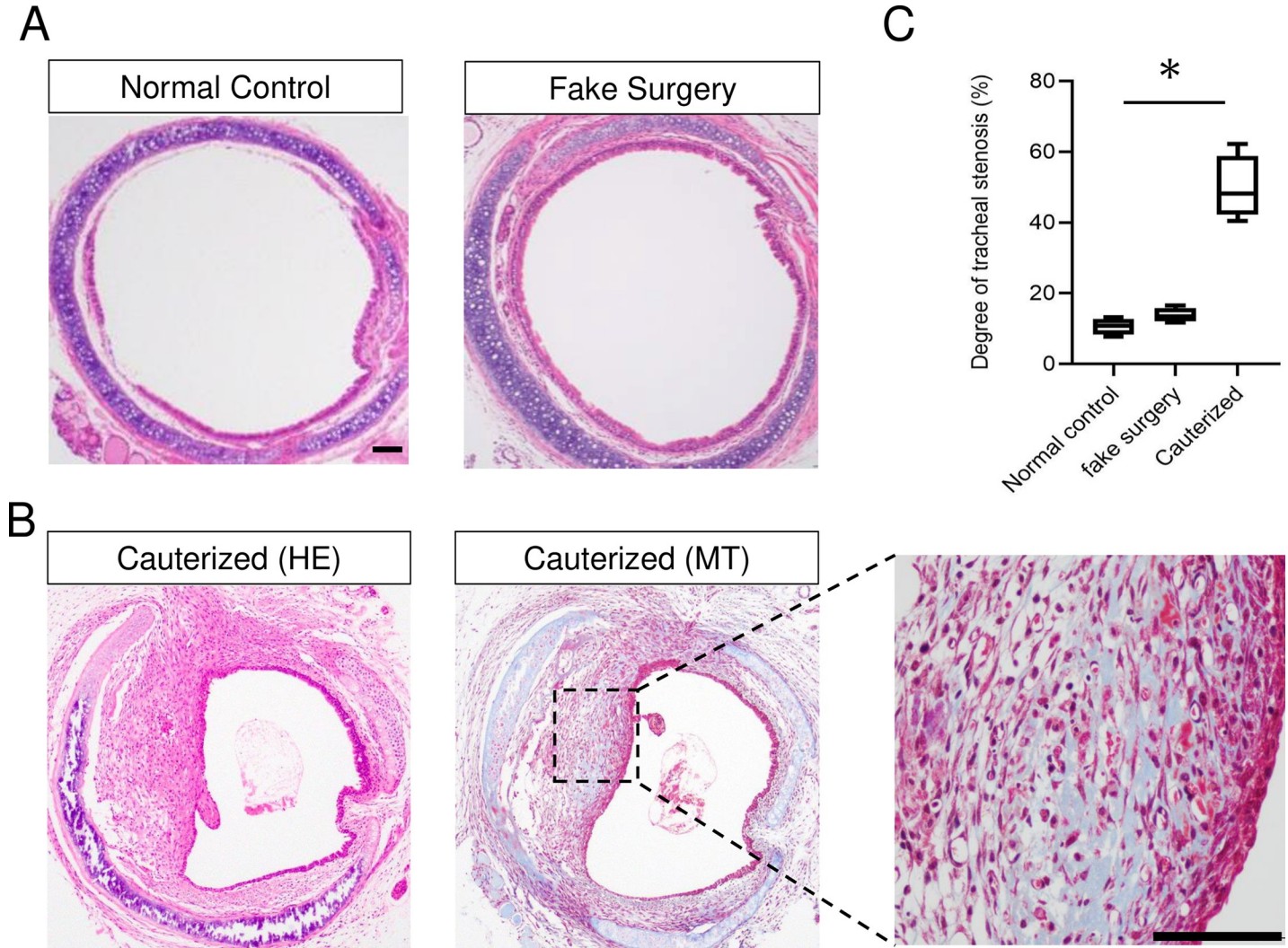

**Fig 2. Axial sections of the trachea.** (A) HE sections of the excised trachea of the normal control group and the fake surgery group (n = 4 for each group). (B) HE and MT-stained sections of the excised trachea of the cauterized group (n = 4). Enlarged image of boxed area shows the fibrosis region. In the cauterized group, extensive fibrosis, thickening, and collagen deposition in the lamina propria were observed. (C) The degree of stenosis. The cauterized group showed a significantly greater degree of stenosis compared with the normal control group. Scale bar = 100 μm, Kruskal–Wallis test, followed by Dunn's multiple comparison test, $^*p < 0.05$. HE: Hematoxylin and eosin. MT: Masson trichrome.

## 2. P-ERK expression in the normal control and cauterized trachea

Tracheal basal cells are stem cells that can self-renew and give rise to ciliated and secretory cells [15]. We therefore sought the positional relationship between tracheal basal cells and p-ERK. The immunohistochemistry results showed that the basal cells were stained with p63. We observed p-ERK1/2 expression in the tracheal basal cells in the normal control and cauterized tracheas (Fig 3A and 3B), which suggests that the p-ERK was located in the tracheal basal cells in both the steady state and the inflammation phase.

## 3. P-ERK expression over time in the cauterized tracheas

The ERK1 phosphorylation rate was significantly increased at 5 min after cauterization compared with the normal control group ($p = 0.045$) and significantly decreased at 90 min after

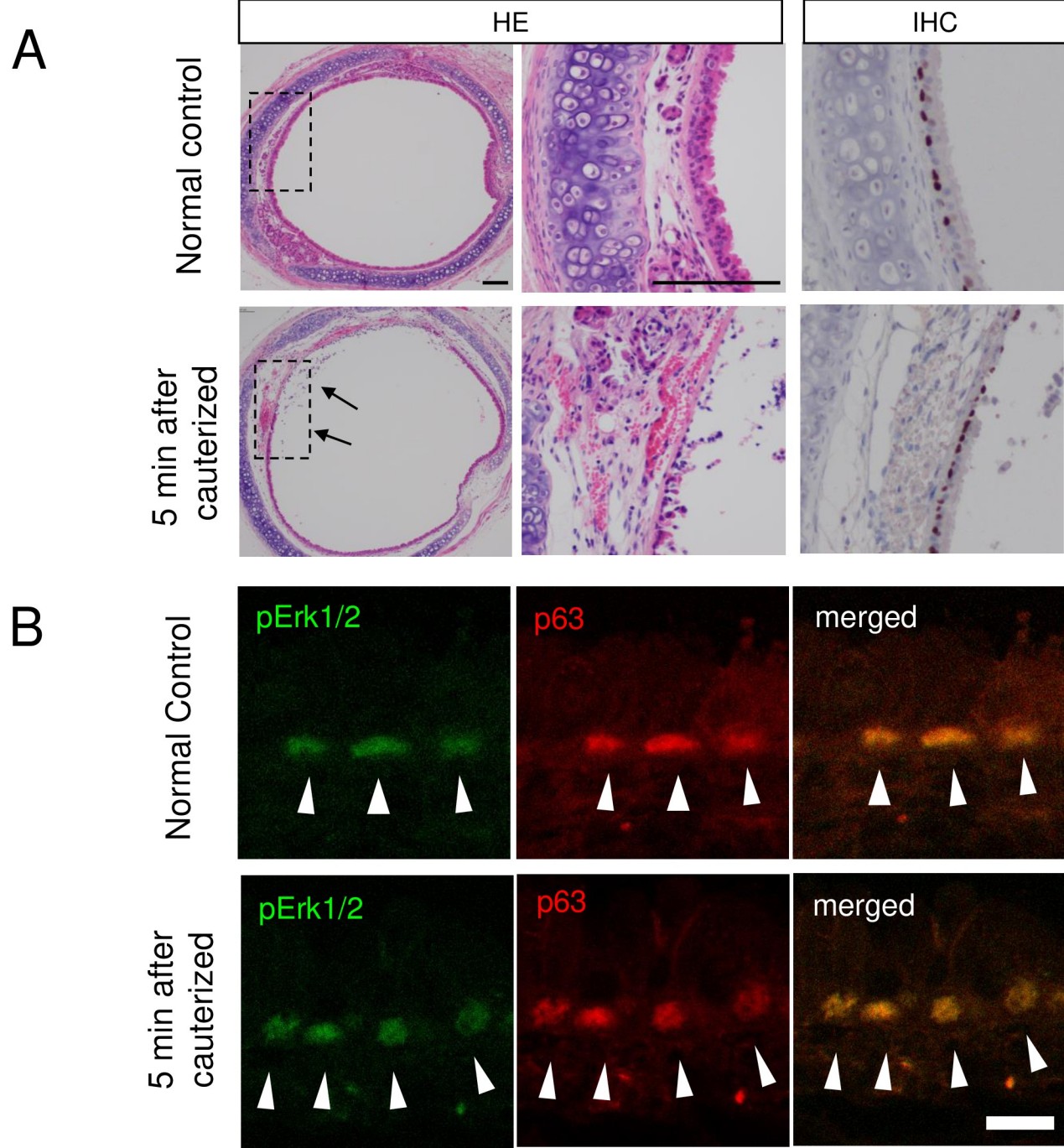

**Fig 3. Local distribution of p-ERK in the normal control trachea and cauterized trachea.** (A) HE and Immunohistochemistry (anti-pERK1/2 antibody) sections of the excised trachea of the normal control and the 5 min after cauterized trachea. We observed p-ERK1/2 expression in the basal cells in the normal control trachea and the cauterized trachea. The injury site of the cauterized trachea showed that the epithelial mucosa was peeling and edematous (arrow heads). Scale bar = 100 μm. (B) Immunofluorescence sections of the excised trachea of the normal control and the 5 min after cauterized trachea. Basal cells were stained with p63 (red). We observed p-ERK1/2 expression (green) in the basal cells in the normal control trachea and cauterized trachea (arrow heads). Scale bar = 10 μm. p-ERK: phosphorylated extracellular signal-regulated kinase.

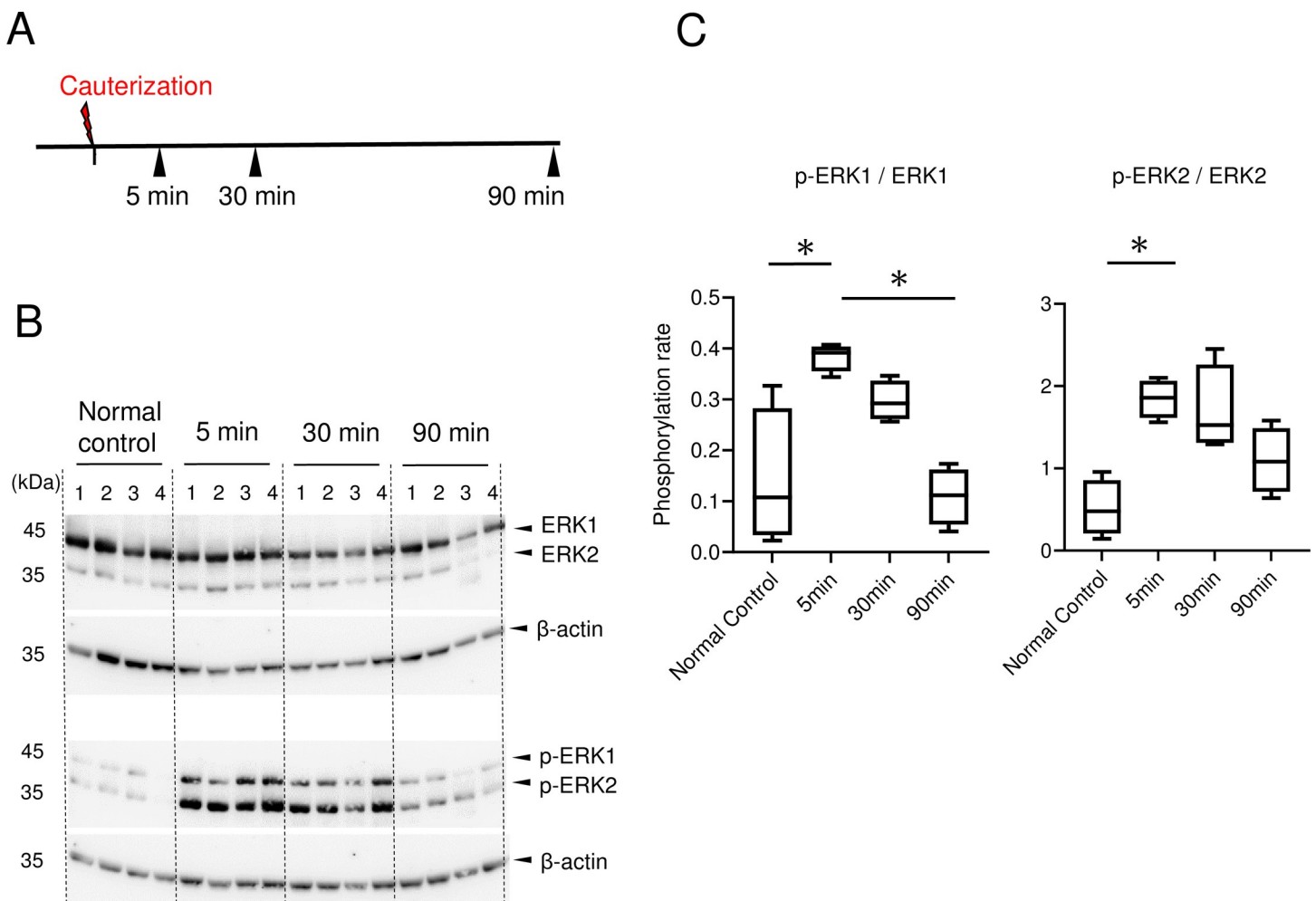

**Fig 4. The p-ERK expression in the cauterized group.** (A) Experimental procedure. (B) Western blot analysis of the anti-ERK1/2 and p-ERK1/2 expression (normal control (non-cauterized) / 5 min, 30 min, and 90 min after cauterization) (n = 4 for each group). (C) The ERK1 phosphorylation rate was significantly increased at 5 min after cauterization compared with the normal control group and significantly decreased at 90 min after cauterization compared with the phosphorylation rate at 5 min. The ERK2 phosphorylation rate was significantly increased at 5 min after cauterization compared with the normal control group. Kruskal–Wallis test followed by Dunn's comparisons test, *$p < 0.05$. p-ERK: phosphorylated extracellular signal-regulated kinase.

cauterization compared with the phosphorylation rate at 5 min ($p = 0.023$). The ERK2 phosphorylation rate was significantly increased at 5 min after cauterization compared with the normal control group ($p = 0.018$) (Fig 4A–4C).

In summary, the tracheal burn injuries caused p-ERK1/2 activation at 5 min after cauterization, after which p-ERK levels decreased over time, suggesting that ERK1/2 phosphorylation could play an important role in the tracheal wound healing process.

## 4. Inhibitory effect of ERK phosphorylation by MEK inhibitor in the normal control tracheas

The ERK1/2 phosphorylation rate at 30 to 120 min after injecting of MEK inhibitor was remarkably decreased (Fig 5A and 5B), which led us to consider that the effect of ERK phosphorylation suppression remained at least 120 min after injecting MEK inhibitor. A previous study reported that MEK inhibitor effectively reduces the basal level of ERK activation for at

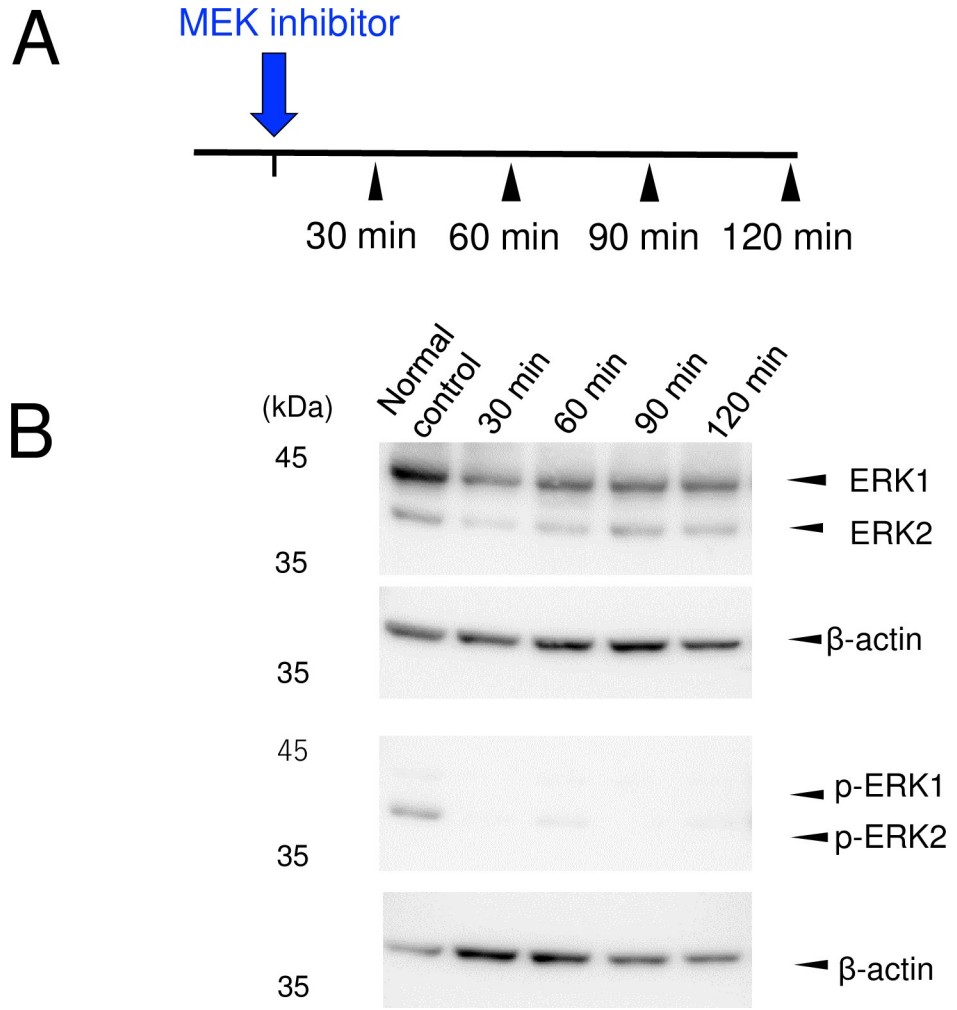

**Fig 5. Inhibitory effect of ERK phosphorylation by MEK inhibitor in the normal control group.** (A) Experimental procedure. The normal (non-cauterized) mice were injected with MEK inhibitor. The tracheas were removed at 30, 60, 90, and 120 min after injection (n = 1 for each group). (B) Western blot analysis of the anti-ERK and p-ERK expression of the excised tracheas. The ERK1/2 phosphorylation rate at 30 to 120 min after injecting MEK inhibitor was remarkably reduced. p-ERK: phosphorylated extracellular signal-regulated kinase.

least 6 h [13]. We therefore administered MEK inhibitor 30 min prior to cauterization to effectively inhibit the ERK phosphorylation of the trachea.

## 5. Inhibitory effect of ERK phosphorylation by MEK inhibitor in cauterized tracheas

The ERK1/2 phosphorylation rate of the MEKi-treated cauterized group was remarkably reduced compared with that of the non-treated cauterized group 5–90 min after cauterization (Fig 6A–6C), which confirms that the inhibitory effect of ERK1/2 phosphorylation by MEK inhibitor injected 30 min prior to cauterization lasts for at least 90 min after cauterization.

## 6. Preventive effect of MEK inhibitor on tracheal stenosis

HE staining revealed that at 7 days after cauterization, the non-inhibitor treatment group showed a narrowed lumen caused by a thickened submucosal layer causing severe stenosis.

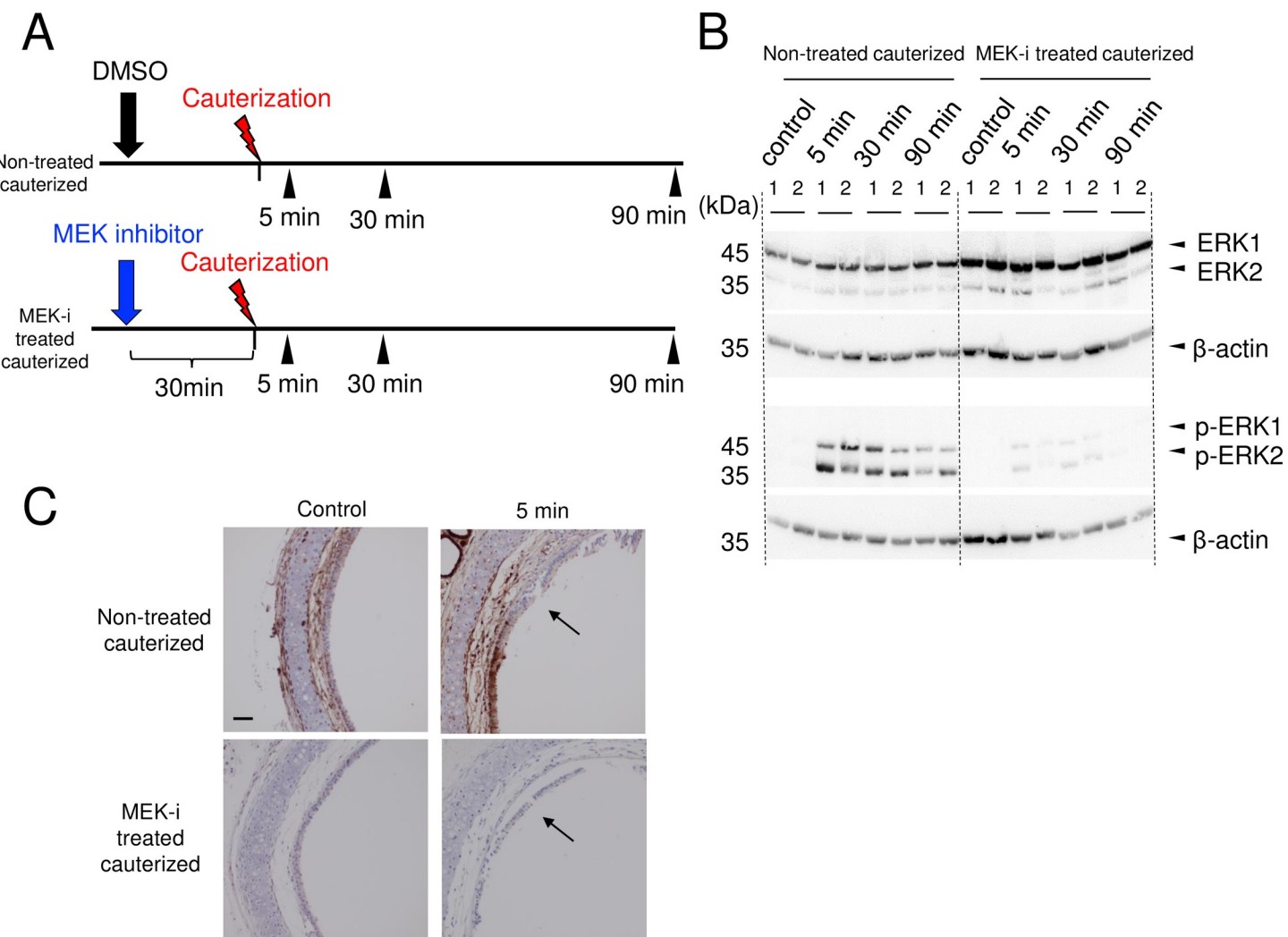

**Fig 6. Inhibitory effect of MEK inhibitor on ERK phosphorylation in the cauterized group.** (A) Experimental procedure. Mice were injected with MEK inhibitor (MEKi-treated cauterized group) or DMSO (non-treated cauterized group) 30 min prior to tracheal cauterization. Tracheas were removed at 5, 30, and 90 min after cauterization (n = 2 for each group). (B) Western blot analysis of the anti-ERK and p-ERK expression of the excised tracheas. The ERK1/2 phosphorylation rate in the MEKi -treated cauterized group was remarkably reduced 5–90 min after cauterization compared with the non-treated cauterized group. (C) IHC sections of the non-treated cauterized and the MEK-i treated cauterized trachea. At both control and 5 min after cauterization, p-ERK1/2 expression in the basal cells were observed in the non-treated trachea but were not observed in the MEK-i treated trachea. Arrow; The injury site of cauterized trachea. DMSO: dimethyl sulfoxide. p-ERK: phosphorylated extracellular signal-regulated kinase.

The median rate of airway stenosis was as follows; non-inhibitor treatment: 51.8%, single treatment: 43.0%, daily treatment: 34.0%. The single treatment group did not show significantly lower rate of airway stenosis compared with the non-inhibitor treatment group ($p$ = 0.424) and the daily treatment group showed a significantly lower rate of airway stenosis compared with the non-inhibitor treatment group ($p$ = 0.024) (Fig 7A–7C).

ERK phosphorylation inhibition through the daily injection of MEK inhibitor successfully suppressed the tracheal stenosis.

## Discussion

In this study, we succeeded in establishing a novel mouse model of tracheal stenosis. Phosphorylated ERK was located in the basal cells in the normal control trachea. Immediately after

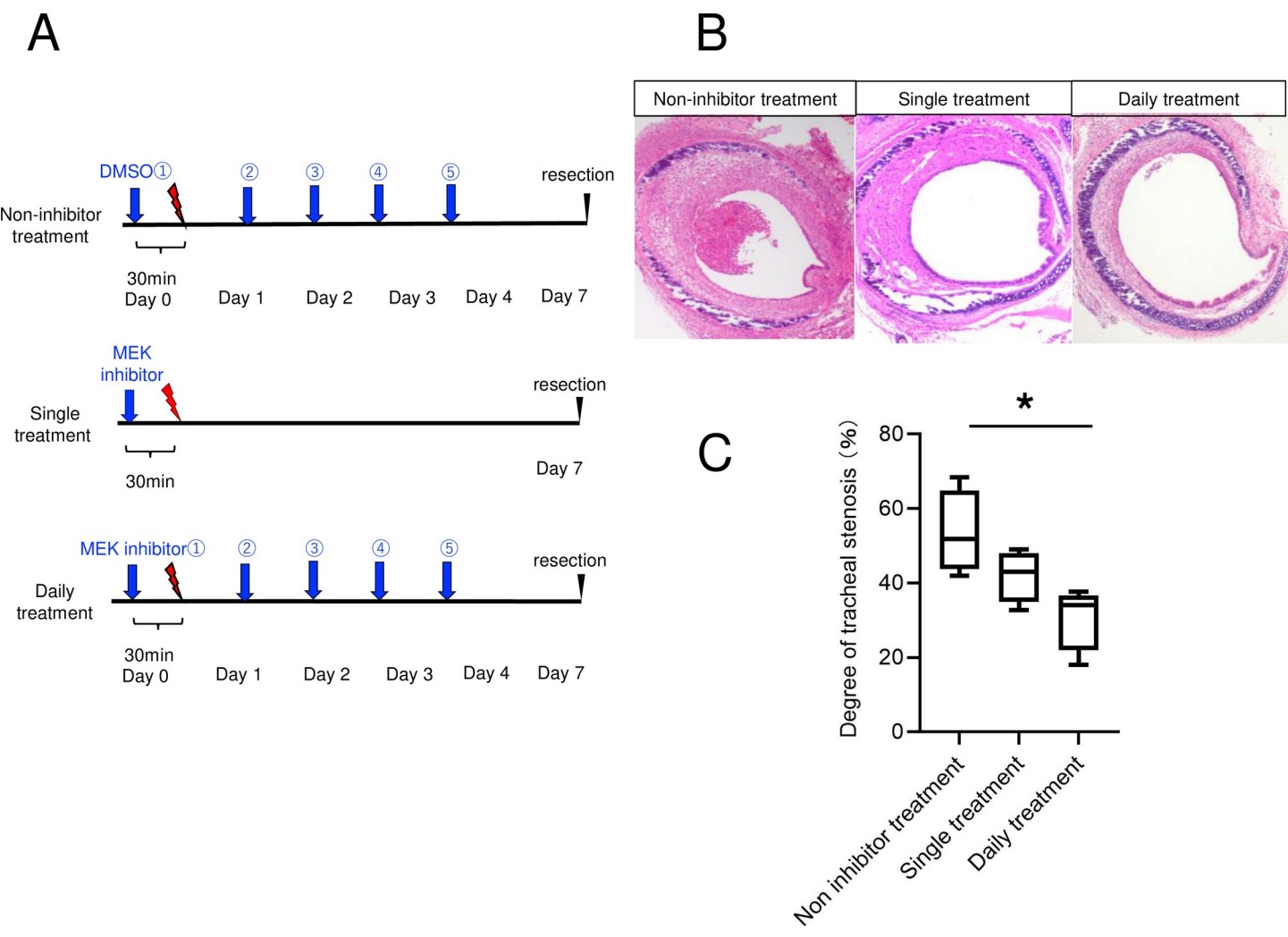

**Fig 7. Preventive effect on tracheal stenosis by the single and daily injection of MEK inhibitor.** (A) Experimental procedure. Non-inhibitor treatment group was injected DMSO 30 min prior to cauterization and 24, 48, 72, and 96 h after cauterization (5 consecutive days). Single treatment group underwent injection with MEK inhibitor 30 min prior to cauterization (n = 4). Daily treatment group underwent injection with MEK inhibitor 30 min prior to cauterization and 24, 48, 72, and 96 h after cauterization (5 consecutive days) (n = 4). Seven days after the cauterization, we excised the tracheas of all groups. (B) HE tracheal sections of non-inhibitor treatment group, single treatment group, and daily treatment group. The non-inhibitor treatment group showed a narrowed lumen caused by a thickened submucosal layer. The single treatment and daily treatment group showed wider tracheal lumen compared with the non-inhibitor treatment group. (C) The degree of stenosis. The single treatment group did not show significantly lower rate of airway stenosis compared with the non-inhibitor treatment group. The daily treatment group showed a significantly lower rate of airway stenosis compared with the non-inhibitor treatment group. Scale bar = 100 μm, Kruskal–Wallis test followed by Dunn's comparisons test, $^*p < 0.05$.

cauterization, the ERK phosphorylation rate in the tracheas was upregulated. Thus, ERK phosphorylation was associated with the formation of stenosis in the wound healing process. We confirmed that the administration of MEK inhibitor at 30 min prior to cauterization effectively suppressed ERK phosphorylation. Furthermore, we histologically confirmed that the administration of MEK inhibitor for 5 consecutive days suppressed the tracheal stenosis, which suggests that the ERK targeting therapy can be a potential strategy for treating tracheal stenosis.

Tracheal basal cells are stem cells that can self-renew and give rise to ciliated and secretory cells during repair following damage to the epithelium. The basal cells of the mouse trachea responded to injury and proliferated rapidly [15]. The ERK/MAPK pathway in the trachea is reported to play a role in the epithelium for progenitor cell maintenance and differentiation.

ERK activation is therefore necessary for the expression of basal cell function [16]. We therefore assumed that the main therapeutic target is ERK in the basal cells that survived after the burn injury.

Previous results have indicated that ERK2 signaling in the dorsal skin of mice is involved in the rapid response of burn healing mechanisms [11]. In our study, the activation of ERKs in the cauterized trachea indicates that ERKs in the basal cells of the trachea play major roles in tracheal burn injuries.

During the proliferative phase, hyperproliferation of fibroblasts and their products disturb the regeneration of normal tracheas by excessive wound healing [17–19]. Given that the tracheal lumen is a restricted space, proliferated fibroblasts in the submucosal layer can induce suffocation. Therefore, the inhibition of the fibroproliferation pathway during the proliferative phase might suppress tracheal stenosis. In this study, tracheal stenosis was not suppressed by a single treatment but was significantly suppressed by daily treatment, which shows that, during the proliferative phase, ERK phosphorylation was inhibited by the repeated administration of MEK inhibitor. To suppress tracheal stenosis, it is important to keep inhibiting ERK activation during the proliferative phase, 2–5 days after the injury. Although we suppose that the inhibitory effect of ERK phosphorylation might increase by the consecutive administration of the MEK inhibitor, there is another possible explanation. SL-327 is a selective inhibitor of MEK1/2 that also targets the activator protein 1(AP-1) at high concentrations [20]. In fact, AP-1 is a well-known crucial signaling pathway in tissue fibrosis and proliferation [21, 22]. We intend to explore this matter further in a future study.

Some mechanisms that also regulate tissue fibrosis include the PI3K/Akt pathway, metalloproteinases, and TGF-β signaling. These are associated with pulmonary fibrosis, which is the result of the progressive accumulation of scar tissue in the lung [23–25]. Furthermore, they are related to the ERK/MAPK pathway [26–28]. These mechanisms could be targets for antifibrotic interventions in tracheal stenosis. In the present study, we could only show the relationship between the ERK pathway and tracheal stenosis. Further research on these pathways and signaling is thus warranted.

In clinical practice, as far as we know, central airway obstruction has not been a target for MEK inhibitors, although they have been employed as therapeutic agents for cancer [10, 29]. One of the merits of MEK inhibitors as therapeutic agents is selectivity. In contrast with most protein kinase inhibitors, MEK inhibitors are non-ATP competitive inhibitors [29] whose binding sites are located in a hydrophobic pocket adjacent to but not overlapping the ATP-binding site [30], which thus accounts for the selectivity of the MEK inhibitor.

This study has a number of limitations. The model does not apply to all clinical types of acquired tracheal stenosis such as injuries due to hot smoke inhalation and intubation. It is also not practical to administer MEK inhibitors to patients prior to injury in clinical practice. In addition, we systemically inhibited ERK phosphorylation. In fact, adverse effects such as rash, diarrhea, and peripheral edema have been reported [31]. The local administration of MEK inhibitor by spraying the injured area could be considered for future studies. Given that the MEK inhibitor inhibited the phosphorylation of both ERK1 and ERK2, we could not investigate the specific function of the individual ERK isoforms. A study using knockout mouse might solve this problem.

Nevertheless, we have for the first time demonstrated that ERK plays an important role in tracheal stenosis formation. Our results suggest that MEK inhibitor could be an alternative novel non-surgical approach in clinical situations such as burn injuries, laser surgery, and recurrent tracheal stenosis. Combination therapies with drugs such as steroids and mitomycin C could be considered in the future [32, 33] and could also be used as a preventive treatment for postoperative restenosis.

## Conclusion

We successfully established a novel mouse model of tracheal stenosis. Tracheal burn injuries cause p-ERK1/2 activation at 5 min after cauterization, indicating that ERK1/2 signaling could play an important role in the tracheal wound healing process. The tracheal stenosis after cauterization was suppressed by 5 consecutive days of administering MEK inhibitor. Our study suggests that the consecutive inhibition of ERK phosphorylation is a potentially useful therapeutic strategy for tracheal stenosis.

## Supporting information

**S1 Fig. Western blot raw image (Fig 4).** Western blot analysis of ERK and its associated β-actin loading control.
(TIF)

**S2 Fig. Western blot raw image (Fig 4).** Western blot analysis of ERK and its associated β-actin loading control.
(TIF)

**S3 Fig. Western blot raw image (Fig 4).** Western blot analysis of p-ERK and its associated β-actin loading control.
(TIF)

**S4 Fig. Western blot raw image (Fig 4).** Western blot analysis of p-ERK and its associated β-actin loading control.
(TIF)

**S5 Fig. Western blot raw image (Fig 5).** Western blot analysis of ERK and its associated β-actin loading control.
(TIF)

**S6 Fig. Western blot raw image (Fig 5).** Western blot analysis of ERK and its associated β-actin loading control.
(TIF)

**S7 Fig. Western blot raw image (Fig 5).** Western blot analysis of p-ERK and its associated β-actin loading control.
(TIF)

**S8 Fig. Western blot raw image (Fig 5).** Western blot analysis of p-ERK and its associated β-actin loading control.
(TIF)

**S9 Fig. Western blot raw image (Fig 6).** Western blot analysis of ERK and its associated β-actin loading control.
(TIF)

**S10 Fig. Western blot raw image (Fig 6).** Western blot analysis of ERK and its associated β-actin loading control.
(TIF)

**S11 Fig. Western blot raw image (Fig 6).** Western blot analysis of p-ERK and its associated β-actin loading control.
(TIF)

**S12 Fig. Western blot raw image (Fig 6).** Western blot analysis of p-ERK and its associated β-actin loading control.
(TIF)

## Author Contributions

**Conceptualization:** Akari Kimura, Koji Araki, Yasushi Satoh, Shogo Endo, Akihiro Shiotani, Taku Yamashita.

**Formal analysis:** Akari Kimura, Koji Araki, Takaomi Kurioka.

**Funding acquisition:** Akari Kimura, Koji Araki.

**Investigation:** Akari Kimura, Koji Araki, Sachiyo Mogi, Kazuko Fujitani.

**Methodology:** Akari Kimura, Koji Araki, Yasushi Satoh, Akihiro Shiotani, Taku Yamashita.

**Project administration:** Akari Kimura, Koji Araki, Yasushi Satoh, Akihiro Shiotani, Taku Yamashita.

**Supervision:** Akihiro Shiotani.

**Writing – original draft:** Akari Kimura, Koji Araki.

**Writing – review & editing:** Yasushi Satoh, Akihiro Shiotani, Taku Yamashita.

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
