## [Decision Letter · Decision Letter 0]

11 May 2021

PONE-D-21-10159

Inhibition of Extracellular signal-regulated kinase pathway suppresses tracheal stenosis in a novel mouse model.

PLOS ONE

Dear Dr. Akari Kimura,

Thank you for submitting your manuscript to PLOS ONE. After careful consideration, we feel that it has merit but does not fully meet PLOS ONE’s publication criteria as it currently stands. Therefore, we invite you to submit a revised version of the manuscript that addresses the points raised during the review process.

We look forward to receiving your revised manuscript.

Kind regards,

Chuen-Mao Yang

Academic Editor

PLOS ONE

Additional Editor Comments:

On the basis of comments from two experts, your manuscript was interested in the field. However, there several questions were needed to be fixed before publication. Could you please take these comments into account to revise the manuscript and re-submitted again.

Journal Requirements:

2. As part of your revisions please address the following items in your revised paper:  (1) the number of animals in each group and how you determined the sample size; (2) the sex and strain of the mice; (3) all anesthetics and analgesics administered to animals during your study (name of drug, dosage, frequency and route of administration); (4) details about humane endpoints for any animals who became severely ill during the study; (5) the rate of mortality during the study and the cause of death (if applicable); (6) the criteria/monitoring parameters used to evaluate the physical health and well-being of the animals. (7) Lastly, please complete and submit the ARRIVE Guidelines checklist (Essential 10 version): https://arriveguidelines.org/resources/author-checklists. Note: this completed checklist will be published as a supporting information file.

Reviewers' comments:

Reviewer's Responses to Questions

**Comments to the Author**

1. Is the manuscript technically sound, and do the data support the conclusions?

Reviewer #1: Partly

Reviewer #2: Partly

2. Has the statistical analysis been performed appropriately and rigorously? 

Reviewer #1: No

Reviewer #2: I Don't Know

3. Have the authors made all data underlying the findings in their manuscript fully available?

Reviewer #1: Yes

Reviewer #2: Yes

4. Is the manuscript presented in an intelligible fashion and written in standard English?

Reviewer #1: No

Reviewer #2: No

5. Review Comments to the Author

Reviewer #1: The authors hypothesized that the inhibition of ERK phosphorylation is a

potential strategy for treating tracheal stenosis. This study aims to inhibit

excessive fibroproliferation during the proliferative phase by inhibiting ERK

phosphorylation. This is an interesting study but several weak points need to be improved in this article.

1. Figures 5B and 6B need to be statistic analysis as figure 4B.

2. In figure 4, besides normal control, the expression of p-Erk/Erk at o min should be displayed.

3. All data values are presented as means ± standard error (SE) in the section of statistics which is not appropriate for nonparametric tests of hypothesis. Please check the mistake through all figure legends.

4. In figure 7, Mann-Whitney U-test was used for statistics but it should be used in 2 groups’ comparisons but not 3 groups. ‘Error bars show SE’ is not for nonparametric tests of hypothesis, here this is a box plot that should depict median and percentile. The baseline (sham control group) of tracheal stenosis should be displayed, which can demonstrate the efficiency of the inhibitor.

5. The background of all Western blots is too smooth, I can not find the margin of membranes. Please show me the raw data and the borders of the membrane should be clear.

6. One dosage of SL-327 can not significantly rescue tracheal stenosis as shown in figure 7E but it seems to significantly reduce the expression of p-Erk. These findings can not rule out the role of other components in the pathogenesis of tracheal stenosis. SL327 is a selective inhibitor for MEK1/2 with IC50 of 0.18 μM/0.22 μM; moreover, SL327 also targets AP-1 with IC5 of 2.03 μM. Thus, tracheal stenosis may be reversed due to AP-1 inhibited by a series of SL-327 administration. In addition, AP-1 is well known as a crucial signaling pathway in tissue fibrosis and cell proliferation. Its role in the prevention of tracheal stenosis should be addressed in this present study.

7. The discussion needs to be improved. Several crucial mechanisms did not discuss. For instance, PI3K/Akt pathway, metalloproteinases, TGF-β signaling, etc. could be pulmonary fibrosis pathways as a target for anti-fibrotic interventions in tracheal stenosis. “MEK inhibitors have already been employed as therapeutic agents for cancer” as the authors mentioned, thus whether central airway obstruction is a target for MEK inhibitor or not. This is also worth discussion.

Reviewer #2: In this manuscript, the authors established a novel mouse model of tracheal stenosis by the usage of tracheostomy and cauterization. The administration of MEK inhibitor seems effectively attenuated the development of tracheal stenosis. The development of experimental model of tracheal stenosis is interesting, but several parts need be further evaluated to assist the results.

1. It needs clearly described where the 3 trachea rings removed. It is addressed that the tracheotomy was performed between the fourth and fifth tracheal rings and where the three removed tracheal rings from?

2. Why is 50 mg/kg of MEK inhibitor used? Any reference?

3. In page 12, the formula to determine the rate of airway stenosis should provide the reference paper.

4. the n value should be provided in each figure legend.

5. In figure 3, what the meaning is to show that the phosphorylation of ERK1/2 is higher in basal cells than other cell types. It's not equal to the situation of ERK1/2 phosphorylation during tracheal stenosis. It needs further assays to show that the more increased phosphorylation of ERK1/2 in basal cells than other cell types during tracheal stenosis.

6. In figure 4, why the time points of 5, 30 and 90 min are chosen. In facts, this may the injury response of cauterized, not the proliferative phase or fibrotic phase of tracheal stenosis.

7. To confirm the result of the Western blot, it would be better if provided the IHF or IHC staining of p-ERK1/2 of figure 4 and 6.Moreover, the basal cells with p-ERK1/2 should be pointed.

8. One more suggestion that the injury site of each IHC data should be pointed.

9. The group comparisons of fake surgery, non inhibitor treatment and treatment groups should be provided.

6. PLOS authors have the option to publish the peer review history of their article (what does this mean?). If published, this will include your full peer review and any attached files.

Reviewer #1: No

Reviewer #2: No

---

## [Author Response · Author response to Decision Letter 0]

27 Jul 2021

Dear Chuen-Mao Yang　　

Academic Editor, 

PLOS ONE

PONE-D-21-10159

July 26, 2021

Thank you very much for the thoughtful and constructive feedback you provided during the review of our manuscript entitled " Inhibition of Extracellular signal-regulated kinase pathway suppresses tracheal stenosis in a novel mouse model."

We have made the suggested revisions in order to better conform to the formatting and content rules of “PLOS ONE”.

We hereby resubmit our manuscript for a secondary evaluation. 

Thank you once again for your consideration of our paper.

Sincerely,

Akari Kimura, MD

Department of Otorhinolaryngology-Head and Neck Surgery, Kitasato University School of Medicine 

1-15-1 Kitasato, Minami, Sagamihara, Kanagawa, 252-0375, Japan

Tel: +81-4-277-8111

Fax: +81-4-2778-9371

Email: akari-k@med.kitasato-u.ac.jp

 

Response to Editor Comments

We have corrected the manuscript to meet PLOS ONE's style requirements including file naming.

2. As part of your revisions please address the following items in your revised paper:

(1) the number of animals in each group and how you determined the sample size; 

The animals were divided into these groups: 

(1) Normal control group, including mice that did not have any surgery [n=8] (n = 4 for histological analysis and n = 4 for Western Blot)

(2) Cauterized group (A), including mice with trachea cauterized by electric scalpel and were sacrificed at seven days after cauterization [n=4] (n = 4 for histological analysis)

(3) Cauterized group (B), including mice with trachea cauterized and were sacrificed at 0, 5, 30, and 90 min after cauterization [n=17] (n = 16 for Western Blot, n=1 for HE, IHC)

(4) Fake surgery group, including mice that did only tracheostomy and were sacrificed at seven days after cauterization [n=4] (n = 4 for histological analysis)

(5) MEK inhibitor treatment group in normal control trachea, including mice with normal trachea which were administrated of SL327 and were sacrificed at 0, 30, 60, 90, and 120 min after the administration. [n=5] (n=5 for Western Blot).

(6) MEK inhibitor / Non-inhibitor treatment in the cauterized group, including mice with cauterized trachea which were administrated of SL327 / DMSO and were sacrificed at 0, 5, 30, and 90 min after the cauterization. [n=20] (n=16 for Western Blot, n=4 for HE, IHC).

(7) Single treatment group including mice with trachea cauterized 30 min before administrated DMSO [n=4] (n =4 for histological analysis)

(8) Daily treatment group including mice with trachea cauterized and administrated SL327 5 days [n=4] (n =4 for histological analysis)

(9) Non inhibitor treatment group, including mice with trachea cauterized and administrated DMSO 5 days [n=4] (n =4 for histological analysis)

The calculation of sample size was based on law of diminishing return, which was called “resource equation” method. By calculation, 4 mice each group were considered as enough sample size.

(2) the sex and strain of the mice; 

We used male C57BL/6J mice between 7 and 9 weeks of age.

(3) all anesthetics and analgesics administered to animals during your study (name of drug, dosage, frequency and route of administration); 

Animals were anesthetized by an intraperitoneal injection of 3 types of mixed anesthetic agents: medetomidine hydrochloride (Domitol®, 0.75 mg/kg), midazolam (Dormicum®, 4 mg/kg), and butorphanol (Vetophale®, 5 mg/kg).

(4) details about humane endpoints for any animals who became severely ill during the study; 

Animals were observed once a day and euthanized when marked wheezing, and significant decrease in body movement were observed. They were treated as deaths and the survival rate was also examined.

(5) the rate of mortality during the study and the cause of death (if applicable); 

The rate of mortality was 9%. The main cause of death was suffocation right after the surgery. Thus, we have observed mice carefully three hours after the surgery, treated warmly, and sucked sputum and blood to prevent acute airway obstruction. 

(6) the criteria/monitoring parameters used to evaluate the physical health and well-being of the animals.；

The animals were maintained at a room temperature (RT) of 24 ±1 °C in a 14 h:10 h light. Animals were observed once a day and monitored for wheezing and significant decrease in body movement.

(7) Lastly, please complete and submit the ARRIVE Guidelines checklist (Essential 10 version): 

We have completed and submitted the ARRIVE Guidelines checklist.

3. PLOS ONE now requires that authors provide the original uncropped and unadjusted images underlying all blot or gel results reported in a submission’s figures or Supporting Information files.

We have submitted all Western blot gel data reported in Supporting Information. 

 

Response to Reviewers

Reviewer #1: 

Thank you very much for reviewing our manuscript and offering valuable advice. We have addressed your comments in a point-by-point manner and have revised the manuscript accordingly. 

1. Figures 5B and 6B need to be statistic analysis as figure 4B.

Answer: Thank you for your suggestion. We consider these experiments just as preliminary experiments confirming the effect of the MEK inhibitor. We used 5 mice (n = 1 for each group) in Figure 5 and 16 mice (n = 2 for each group) in Figure 6. In keeping with humane consideration, increasing the number of sacrificed mice was deemed inappropriate. Thus, we could not perform statistical analysis.

2. In figure 4, besides normal control, the expression of p-Erk/Erk at o min should be displayed.

Answer: You have raised an important point. However, it takes a few minutes to remove the trachea. Thus, we defined “5 min after cauterization” as “the time just after cauterization”. Furthermore, we defined “Normal Control” to mean “0 min before cauterization”.

3. All data values are presented as means ± standard error (SE) in the section of statistics which is not appropriate for nonparametric tests of hypothesis. Please check the mistake through all figure legends.

Answer: Thank you for this important suggestion. We agree with your assessment and have corrected the data expression.

We used a box plot for all data in our manuscript. [Fig 2C,4C, 6B, and 7C]

4. In figure 7, Mann-Whitney U-test was used for statistics but it should be used in 2 groups’ comparisons but not 3 groups. ‘Error bars show SE’ is not for nonparametric tests of hypothesis, here this is a box plot that should depict median and percentile. The baseline (sham control group) of tracheal stenosis should be displayed, which can demonstrate the efficiency of the inhibitor.

Answer: Thank you for your suggestion. 

We have performed additional experiments which include “fake surgery group (only tracheostomy, no cauterization)” (n = 4) and a “non-inhibitor treatment group (trachel cauterized and administered with DMSO for 5days)” (n = 4). We removed the tracheas in both groups 7 days after surgery. 

At first, we assigned 12 mice to 3 groups “normal control group, fake surgery group, and cauterized group” (n=4 for each group). 

Furthermore, as treatment groups, we assigned 12 mice to 3 groups “non-inhibitor treatment group, single treatment group, and daily treatment group) (n=4 for each group). 

We analyzed the intergroup comparisons using the non-parametric Kruskal–Wallis test, followed by Dunn’s multiple comparison test, with the statistical significance level set at p values < 0.05. [Fig 2A-C, Fig 7A-C] We used plots to present median and percentile

5. The background of all Western blots is too smooth, I cannot find the margin of membranes. Please show me the raw data and the borders of the membrane should be clear.

Answer: The raw data of Western blots (Fig. 4B,5B, and 6B) are presented in Supporting information [S1 Fig]. 

6. One dosage of SL-327 can not significantly rescue tracheal stenosis as shown in figure 7E but it seems to significantly reduce the expression of p-Erk. These findings can not rule out the role of other components in the pathogenesis of tracheal stenosis. SL327 is a selective inhibitor for MEK1/2 with IC50 of 0.18 μM/0.22 μM; moreover, SL327 also targets AP-1 with IC5 of 2.03 μM. Thus, tracheal stenosis may be reversed due to AP-1 inhibited by a series of SL-327 administration. In addition, AP-1 is well known as a crucial signaling pathway in tissue fibrosis and cell proliferation. Its role in the prevention of tracheal stenosis should be addressed in this present study.

Answer: Thank you for this valuable suggestion. There is another possible explanation. SL-327 is a selective inhibitor of MEK1/2 with an IC50 of 0.18 μM/0.22 μM that also targets the activator protein 1(AP-1) with an IC50 of 2.03 μM at high concentrations. In fact, AP-1 is a well-known crucial signaling pathway in tissue fibrosis and proliferation. AP-1 is downregulated at early time points and upregulated at later time points in normal reepithelialization during wound healing. This indicates that MEK and AP-1 have opposite effects. We intend to explore this matter further in a future study.

We have revised the text in the Discussion section [P24, L400-405] and have added the following text, as per your comment: 

“Although we suppose that the inhibitory effect of ERK phosphorylation might increase by the consecutive administration of the MEK inhibitor, there is another possible explanation. SL-327 is a selective inhibitor of MEK1/2 that also targets the activator protein 1(AP-1) at high concentrations [20]. In fact, AP-1 is a well-known crucial signaling pathway in tissue fibrosis and proliferation [21] [22]. We intend to explore this matter further in a future study.

7. The discussion needs to be improved. Several crucial mechanisms did not discuss. For instance, PI3K/Akt pathway, metalloproteinases, TGF-β signaling, etc. could be pulmonary fibrosis pathways as a target for anti-fibrotic interventions in tracheal stenosis. “MEK inhibitors have already been employed as therapeutic agents for cancer” as the authors mentioned, thus whether central airway obstruction is a target for MEK inhibitor or not. This is also worth discussion.

Answer: We have revised the text according to your suggestions. We have added the following text [P24, L406- P25, L412]:

“Some mechanisms that also regulate tissue fibrosis include the PI3K/Akt pathway, metalloproteinases, and TGF-β signaling. These are associated with pulmonary fibrosis, which is the result of the progressive accumulation of scar tissue in the lung [23-25]. Furthermore, they are related to the ERK/MAPK pathway [26-28]. These mechanisms could be targets for anti-fibrotic interventions in tracheal stenosis. In the present study, we could only show the relationship between the ERK pathway and tracheal stenosis. Further research on these pathways and signaling is thus warranted.”

 

Reviewer #2:

Thank you very much for reviewing our manuscript and offering valuable advice. We have addressed your comments in a point-by-point manner and have revised the manuscript accordingly. 

 1. It needs clearly described where the 3 trachea rings removed. It is addressed that the tracheotomy was performed between the fourth and fifth tracheal rings and where the three removed tracheal rings from?

Answer: You have raised an important question. We performed tracheostomy between the fourth and fifth tracheal rings and cauterized the third and fourth rings. Then, we removed the tracheal rings and analyzed at during the first and second rings where showed most severe tracheal stenosis. We have added the following information in below in the Figure 1 legend [P8, L109-113]:

(A) Tracheostomy was performed between the fourth and fifth tracheal rings (blue area). The anterior tracheal wall above the tracheostoma was cauterized using an electric scalpel (red area). (B) Trachea after tracheostomy. (C) Trachea after cauterization. Green box: removed area, yellow lines: sliced sections.

2. Why is 50 mg/kg of MEK inhibitor used? Any reference?

Answer: We have added the following reference accordingly: “Transient Blockade of ERK Phosphorylation in the Critical Period Causes Autistic Phenotypes as an Adult in Mice. Sci Rep. 2015;5:10252” [P27, L 474-476]

3. In page 12, the formula to determine the rate of airway stenosis should provide the reference paper.

Answer: We have determined the rate of tracheal airway stenosis as described in ”Gene Therapy of c-myc Suppressor FUSE Binding　Protein-Interacting Repressor by Sendai Virus Delivery Prevents Tracheal Stenosis” (PLOS ONE,2015) [P28, L477-479]. We have added this reference to the reference list.

4. the n value should be provided in each figure legend.

Answer: We have made the suggested revisions accordingly. 

5. In figure 3, what the meaning is to show that the phosphorylation of ERK1/2 is higher in basal cells than other cell types. It's not equal to the situation of ERK1/2 phosphorylation during tracheal stenosis. It needs further assays to show that the more increased phosphorylation of ERK1/2 in basal cells than other cell types during tracheal stenosis.

Answer: You have raised an important point. However, the ERK phosphorylation rate is significantly increased at 5 min and decreased 90 min after cauterization [Fig 4]. This suggests that p-ERK expression is lower during the proliferative phase and during tracheal stenosis. For this reason, we presented the phosphorylation of ERK1/2 in basal cells at 5 min after cauterization. [Fig3]

6. In figure 4, why the time points of 5, 30 and 90 min are chosen. In facts, this may the injury response of cauterized, not the proliferative phase or fibrotic phase of tracheal stenosis.

Answer: In this study, we aimed to clarify the effect of the suppression of ERK phosphorylation during early inflammation, not during the proliferative or fibrotic phase of tracheal stenosis. This justifies the examination of the activation of ERK at early time points after cauterization. 

7. To confirm the result of the Western blot, it would be better if provided the IHF or IHC staining of p-ERK1/2 of figure 4 and 6.　Moreover, the basal cells with p-ERK1/2 should be pointed.

Answer: Thank you for your suggestion. We have added new figures that include the IHC staining of p-ERK1/2 in tracheas. [Fig 3AB, Fig6C]

8. One more suggestion that the injury site of each IHC data should be pointed.

Answer: Thank you for your suggestion. We have added figures which present the injury site of each HE and IHC data. 

The injury sites which were pointed by arrows have showed that the epithelial mucosa was peeling and edematous. [Fig 3AB,Fig 6C]

9. The group comparisons of fake surgery, non inhibitor treatment and treatment groups should be provided.

Answer: Thank you for providing your assessment. 

According to your suggestion, we have performed additional experiments that include a “fake surgery group” (n = 4) and “non-inhibitor treatment group” (n = 4). We removed tracheas in both groups 7 days after surgery.

We assigned 12 mice to 3 groups “normal control group, fake surgery group, and cauterized group” (n=4 for each group). 

Furthermore, as treatment groups, we assigned 12 mice to 3 groups “non-inhibitor treatment group, single treatment group, and daily treatment group) (n=4 for each group). 

We analyzed the intergroup comparisons using the non-parametric Kruskal–Wallis test, followed by Dunn’s multiple comparison test, with the statistical significance level set at p values < 0.05. [Fig 2A-C, Fig 7A-C]

---

## [Editor Report · Decision Letter 1]

30 Jul 2021

Inhibition of Extracellular signal-regulated kinase pathway suppresses tracheal stenosis in a novel mouse model.

PONE-D-21-10159R1

Dear Dr. Akari Kimura,

We’re pleased to inform you that your manuscript has been judged scientifically suitable for publication and will be formally accepted for publication once it meets all outstanding technical requirements.

Kind regards,

Chuen-Mao Yang

Academic Editor

PLOS ONE
---

## [Editor Report · Acceptance letter]

20 Sep 2021

PONE-D-21-10159R1 

Inhibition of Extracellular signal-regulated kinase pathway suppresses tracheal stenosis in a novel mouse model. 

Dear Dr. Araki:

I'm pleased to inform you that your manuscript has been deemed suitable for publication in PLOS ONE. Congratulations! Your manuscript is now with our production department. 

Kind regards, 

on behalf of

Dr. Chuen-Mao Yang 

Academic Editor

PLOS ONE